# Safety outcomes of ticagrelor among patients with STE-ACS post streptokinase therapy-a retrospective observational study

Phornpaka Ueapornpanith[1], Boonyanuch Buranakiti[2], Thanyalak Chotayaporn[3], Arintaya Phrommintikul[4], Voratima Yoodee[1]¤ *

**1** Faculty of Pharmacy, Chiang Mai University, Chiang Mai, Thailand, **2** Pharmacy Department, Nakornping Hospital, Chiang Mai, Thailand, **3** Department of Internal Medicine, Nakornping Hospital, Chiang Mai, Thailand, **4** Faculty of Medicine, Department of Internal Medicine, Division of Cardiology, Chiang Mai University, Chiang Mai, Thailand

¤ Current address: Faculty of Pharmacy, Department of Pharmaceutical Care, Pharmaceutical Care Training Center (PCTC), Chiang Mai University, Chiang Mai, Thailand

* voratima.silavanich@gmail.com

**Data Availability Statement:** Data cannot be shared publicly because data belong to third party (Maharaj Nakorn Chiang Mai and Nakornping

## Abstract

From the restriction of access to primary percutaneous coronary intervention, about 46% of patients with ST-elevation acute coronary syndrome (STE-ACS) received fibrinolytic therapy as a reperfusion strategy; streptokinase is frequently used in Thailand. Despite the guidelines recommending potent $P2Y_{12}$ inhibitors among these patients, the data are limited, especially among patients with STE-ACS post streptokinase therapy. The study was proposed to describe factors for $P2Y_{12}$ inhibitors selection and evaluate outcomes of pharmacoinvasively treated STE-ACS receiving ticagrelor compared with clopidogrel in Thailand. We performed a retrospective observational study of patients with STE-ACS post streptokinase therapy followed by percutaneous coronary intervention (PCI) with coronary stent placement and receiving ticagrelor or clopidogrel as $P2Y_{12}$ inhibitor treatment from January 2017 to June 2021. The primary outcomes described factors for $P2Y_{12}$ inhibitor selection and evaluated safety outcomes with inverse probability weight (IPW) adjustment. The secondary outcome was a composite of all-cause death, myocardial infarction and stroke. The median time from streptokinase therapy to initiating ticagrelor in the switch group was 25.7 (IQR, 1.9–4.4) hours. The factors related to switching from clopidogrel to ticagrelor included young age, history of coronary artery disease (CAD), dose of streptokinase and use of intravascular imaging. Any bleeding events occurred among 83 patients (41.71%) in the switch group and 83 patients (41.09%) in the no switch group (adjusted HR 1.04, 95% CI 0.75–1.44; p = 0.826). The composite of efficacy outcomes occurred in 6 patients in the switch group (3.02%) and 12 patients (5.94%) in the no switch group (adjusted HR 0.57, 95% CI 0.21–1.57; p = 0.279). Conclusion: In real practice, ticagrelor switching among patients with STE-ACS post streptokinase therapy did not differ regarding safety outcomes and composite of efficacy outcomes compared with clopidogrel.

hospital). Data are available from the Operation and data management center (contact via oc. med@cmu.ac.th and Tell +66 53 999 200 Ext. 1174 for Maharaj Nakorn Chiang Mai and Nakornping hospital, respectively) for researchers who meet the criteria for access to confidential data.

**Funding:** The authors received no specific funding for this work.

**Competing interests:** The authors have declared that no competing interests exist.

## Introduction

Currently, PPCI is the preferred reperfusion strategy among patients with ST-elevation acute coronary syndrome (STE-ACS) to decrease mortality outcomes. In some circumstances, PPCI cannot be an immediate option; fibrinolysis is required as the reperfusion strategy [1–3]. The benefit of dual antiplatelet among patients with STE-ACS post fibrinolytic therapy has been established [4]. Randomized trials demonstrated potent $P2Y_{12}$ inhibitors as ticagrelor and prasugrel significantly decreasing the composite efficacy outcome compared with clopidogrel; however, the trials excluded patients receiving fibrinolysis at 24 to 48 hours before enrolling [5,6]. As a result, the European Society of Cardiology (ESC) guidelines for the management of acute myocardial infarction in patients presenting with ST-segment elevation recommend considering switching clopidogrel to potent $P2Y_{12}$ inhibitors 48 hours after fibrinolysis among patients undergoing PCI [1].

In the research, Ticagrelor vs. Clopidogrel After Fibrinolytic Therapy in Patients With ST-Elevation Myocardial Infarction: a Randomized Clinical Trial (TREAT trial) [7], ticagrelor administration after fibrinolytic therapy 11.4 hours was proved noninferior to clopidogrel in the Thrombolysis in Myocardial Infarction (TIMI) major bleeding criteria. Another study by Welsh RC et al. [8] found that switching to ticagrelor at 9.9 hours after fibrinolysis pharma-coinvasive reperfusion did not differ regarding major bleeding. However, the fibrinolytic therapy in the TREAT trial [7] barely included streptokinase (5.7% in the ticagrelor group), and the study by Welsh RC et al. [8] consisted of tenecteplase only.

In Thailand, access to PPCI can be delayed or made impossible by geographic and economic hurdles, such as long distances to PCI Centers, difficulties with traffic and transportation, PCI Centers not being open overnight or over weekends or being unavailable. From the Thai Registry in Acute Coronary Syndrome (TRACS) [9], 55% presented STE-ACS; thrombolysis was provided to 42.6%, and PPCI was performed to 24.7%. Consequently, fibrinolysis remains the main reperfusion strategy for patients with STE-ACS in Thailand. Moreover, fibrinolytic drug selection follows the National List of Essential Medicines and national health insurance criteria, leading to 96% of fibrinolytic therapy involving streptokinase [10]. Therefore, the Safety outcomes of $P2Y_{12}$ inhibitors among patients with STE-ACS post streptokinase therapy study (SP-SK study) was investigated to describe factors for $P2Y_{12}$ inhibitor selection. Further, safety outcomes of ticagrelor versus clopidogrel were evaluated in real clinical practice in Thailand.

## Materials and methods

### Study design

The retrospective observational study consisted of two PCI Centers including Maharaj Nakorn Chiang Mai Hospital, affiliated with Chiang Mai University and Nakornping Hospital, a tertiary care center, responding to one part of the regional health system in northern Thailand. This study was conducted between January 2017 and June 2021, and data were collected from hospital electronic databases. All data recordings were fully anonymized. The study was approved by the ethics committee of the Faculty of Medicine, Chiang Mai University (NONE-2564-08440) and Nakornping Hospital (098/64), which waived the informed consent requirement.

### Study population

Patients with STE-ACS, identified using the International Classification of Diseases 10th Revision (ICD-10: I21.00-I21.3 and I21.9) [11], and post streptokinase therapy were admitted to

the study centers. Inclusion criteria comprised patients undergoing PCI with coronary stent placement after fibrinolysis, identified by the International Classification of Diseases 9th Revision, Clinical Modification (ICD-9-CM: 00.40–00.44 and 00.45–00.49) [12], and at least one was followed up at the outpatient department. Patients with incomplete data or receiving long term oral anticoagulants were excluded. The switch group was defined as patients receiving at least one dose of ticagrelor before hospital discharge, and the no switch group was defined as patients with continued clopidogrel for P2Y$_{12}$ inhibitor treatment.

## Outcomes

The primary outcomes included factors for P2Y$_{12}$ inhibitor selection and safety outcomes as any bleeding events. In addition, the severity of bleeding event was classified by Bleeding Academic Research Consortium (BARC) criteria. The secondary outcome was a composite of all-cause death, myocardial infarction (MI) and stroke within 30 days from streptokinase administration. All outcomes were cross-checked with ICD-10 and ICD-9-CM. Among patients were censored when the outcomes occurred or completed the follow-up time.

## Statistical analysis

Baseline characteristics were described as median (IQR), mean (±SD) for continuous variables or count and percentage for categorical variables. Differences between groups (switch versus no switch groups) were tested using *t*-test or Mann-Whitney U test for continuous variables and Fisher's exact test for categorical variables, respectively. The safety and efficacy outcomes were evaluated by time to first event. To decrease selection bias, outcomes were computed using Cox proportional hazard model with inverse probability weighting (IPW) adjustment. The standardized differences of less than 10% were accepted as negligible imbalance in baseline characteristics [13]. The factors for P2Y$_{12}$ inhibitor selection were analyzed using logistic regression. Univariate and multivariate regression analyses were performed to determine the association between baseline covariates and P2Y$_{12}$ inhibitor switching. Multivariate regression analyses for P2Y$_{12}$ inhibitor switching was constructed using a p-value less than 0.05 on the univariate analysis for entry. All statistical significance was set at p-value less than 0.05.

## Results

### Study population

Of the 401 patients with STE-ACS receiving a pharmacoinvasive reperfusion strategy between 2017 and 2021, 202 were sustained on clopidogrel (no switch group), while 199 were switched in-hospital to ticagrelor (switch group). A flow chart diagram of the study is shown in Fig 1. Mean follow-up time was 21.7±10.0 days. The mean age was 62.5±10.5 years. In the switch group, participants were significantly younger than those in the no switch group (61.2±10.0 and 63.7±10.9, respectively, p = 0.017). Other baseline characteristics differed between the two groups including mean body weight (60.3±11.6 and 57.4±12.2, respectively, p = 0.015), a history of cardiovascular disease (41.7% and 19.8%, respectively, p<0.001), mean estimated glomerular filtration rate (eGFR) (78.7±23.5 and 72.1±23.9, respectively, p = 0.005), dose of streptokinase, PCI access site as femoral access and use of intravascular imaging. However, after adjusting IPW, the baseline characteristics were well balanced. The adjusted IPW is presented in S2 Table. The median time from symptoms onset to streptokinase administration was 2.9 (IQR, 6.9–26.3) hours. Among the switch group patients, ticagrelor was initiated at 25.7 (IQR, 14.6–40.8) hours after streptokinase, and the switching was mostly established post PCI. The median hospital length of stay was three (IQR, 2–4) days. A total of 395 patients

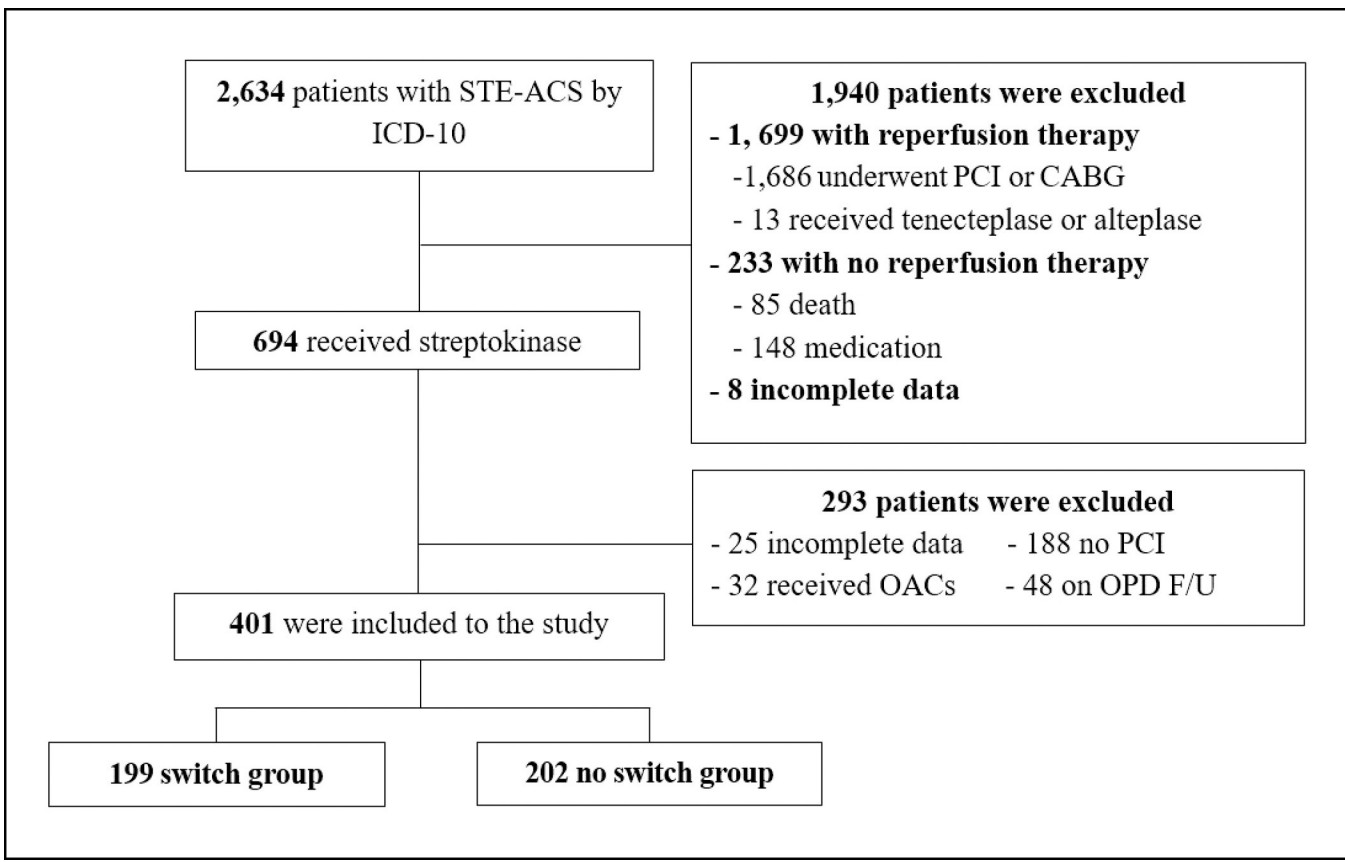

**Fig 1. Study flow chart.** STE-ACS, ST-elevation acute coronary syndrome; ICD-10, International Classification of Diseases 10th Revision; PCI, percutaneous coronary intervention; CABG, coronary artery bypass graft OACs, oral anticoagulants; OPD, outpatient department; F/U, follow-up.

(98.5%) received clopidogrel, mostly 300 mg, in the beginning. In the switch group 148 patients (74.37%) received ticagrelor 180 mg loading. The patient characteristics are presented in Tables 1 and S1.

## Primary outcome

From univariate analysis, age, medical history of CAD, receiving low dose streptokinase, body weight and use of intravascular imaging were found to be statistically significant at $p<0.05$. Therefore, those variables were chosen to enter to the multiple regression analysis in the second step. However, after multivariate regression analysis, the factors that significantly affected $P2Y_{12}$ inhibitors selection included age, medical history of CAD, receiving streptokinase of 0.75 million units and use of intravascular imaging. $P2Y_{12}$ inhibitors switching was preferred when patients met the factors above; and thus, among patients at younger age. More details are presented in Table 2.

The primary safety outcome (any bleeding events) is presented in Table 3. Bleeding events occurred in the switch group at 83 patients (41.71%) and for the no switch group at 83 patients (41.09%). No significant differences were noted in any bleeding events between the switch and no switch groups (adjusted HR 1.04, 95% CI 0.74–1.45; p = 0.832). Kaplan-Meier curves of any bleeding events are presented in Fig 2. In the switch group, 41% of bleeding events occurred before initiating ticagrelor. Major bleeding events as BARC type 3 or 5 occurred among 11 patients (5.53%) in the switch group and 13 patients (6.44%) in the no switch group (adjusted

**Table 1. Baseline characteristics.**

| Characteristics | All (n = 401) | Switch (n = 199) | No switch (n = 202) | p-value [1] |
|---|---|---|---|---|
| Age, mean±SD | 62.5±10.5 | 61.2±10.0 | 63.7±10.9 | 0.017 |
| Female, n (%) | 147 (36.7) | 67 (33.7) | 80 (39.6) | 0.254 |
| Body weight, mean±SD | 58.8±11.9 | 60.3±11.6 | 57.4±12.2 | 0.015 |
| ≥60 kg, n (%) | 189 (47.1) | 106 (53.3) | 83 (41.1) | 0.016 |
| hemoglobin, g/dL, mean±SD | 13.0±2.0 | 13.2±1.9 | 12.2±2.0 | 0.057 |
| hemoglobin <11 g/dL, n (%) | 58 (14.5) | 28 (14.1) | 30 (14.9) | 0.468 |
| Medical history, n (%) | | | | |
| HT | 166 (41.4) | 78 (39.2) | 88 (43.6) | 0.417 |
| DLP | 170 (42.4) | 88 (44.2) | 82 (40.6) | 0.481 |
| DM | 71 (17.7) | 40 (20.1) | 31 (15.3) | 0.240 |
| CAD | 123 (30.7) | 83 (41.7) | 40 (19.8) | <0.001 |
| Stroke, TIA | 7 (1.8) | 4 (2.0) | 3 (1.5) | 0.492 |
| eGFR, mL/min/1.73 $m^2$, mean±SD | 75.4±23.9 | 78.7±23.5 | 72.1±23.9 | 0.005 |
| eGFR <30 mL/min/1.73 $m^2$, n (%) | 19 (4.7) | 7 (3.5) | 12 (5.9) | 0.348 |
| Time report | | | | |
| symptoms onset to streptokinase, h, median (IQR) | 2.9 (1.9–4.5) | 2.8 (1.9–4.4) | 3.0 (1.9–4.6) | 0.656 |
| streptokinase to ticagrelor, h, median (IQR) | - | 25.7 (14.6–40.8) | - | - |
| streptokinase to PCI, h, median (IQR) | 15.7 (6.9–26.3) | 16.7 (7.2–26.9) | 14.6 (5.8–26.3) | 0.615 |
| Anticoagulant administration Pre-PCI, n (%) | 207 (51.6) | 99 (49.7) | 108 (53.5) | 0.485 |
| PCI strategy, n (%) | | | | |
| rescue PCI | 89 (22.2) | 37 (18.6) | 52 (25.7) | 0.093 |
| routine PCI | 312 (77.8) | 162 (81.4) | 150 (74.3) | |

[1] without IPW adjusted; SD, standard deviation; HT, hypertension; DLP, dyslipidemia; DM, diabetes mellitus; CAD; cardiovascular disease; TIA, transient ischemic attack; eGFR, estimated glomerular filtration rate; IQR, interquartile range; PCI, percutaneous coronary intervention; IQR, interquartile range.

HR 1.31, 95% CI 0.55–3.10; p = 0.545). Almost all the bleeding events met BARC type 1 or 2 criteria, 72 patients (36.18%) and 70 patients (34.65%) in the switch group and the no switch group, respectively.

Totally, 56.6% of bleeding events were associated with the PCI access site with a tendency to non- major bleeding; more details of bleeding site are presented in S1 Table. Moreover, a total of 75.9% and 63.9% of bleeding events in the switch group and the no switch group, respectively, occurred within 24 hours after streptokinase therapy (Fig 3). Fig 4 shows bleeding events according to time from receiving streptokinase to initial administration of ticagrelor, including only events occurring after initiating ticagrelor. More bleeding events were exhibited when ticagrelor was initiated before 48 hours (Fig 4A), and BARC type 3 or 5 showed a tendency toward increasing when initiating ticagrelor within 24 hours from streptokinase therapy (Fig 4B).

## Secondary outcome

The composite of efficacy outcome of all-cause death, MI, or stroke totaled 6 events (3.02%) in the switch group and 12 events (5.94%) in the no switch group. After adjusting IPW, the switch group was associated with a nonsignificant difference in composite efficacy outcome (adjusted HR 0.57, 95% CI 0.21–1.57; p = 0.279). Also, no differences were noted in individual composite outcome. Efficacy outcomes are presented in Table 3.

**Table 2. Result of logistic regression adjusted for variables associated with P2Y$_{12}$ inhibitors switching.**

| Characteristics | Univariable OR (95% CI) | p-value | Multivariable OR (95% CI) | p-value |
|---|---|---|---|---|
| Age | 0.98 (0.96–0.99) | 0.018 | 0.97 (0.95–0.99) | 0.029 |
| Female | 0.77 (0.52–1.16) | 0.218 | NA | NA |
| Body weight | 1.02 (1.00–1.04) | 0.016 | 1.01 (0.99–1.03) | 0.252 |
| hemoglobin [a] | | | | |
| ≥11 g/dL | reference | - | - | - |
| <11 g/dL | 0.94 (0.54–1.64) | 0.824 | NA | NA |
| Medical history | | | | |
| HT | 0.84 (0.56–1.24) | 0.375 | NA | NA |
| DLP | 1.16 (0.78–1.72) | 0.463 | NA | NA |
| DM | 1.34 (0.83–2.33) | 0.214 | NA | NA |
| CAD | 2.90 (1.85–4.53) | <0.001 | 2.36 (1.41–3.94) | 0.001 |
| Stroke, TIA | 1.36 (0.30–6.16) | 0.689 | NA | NA |
| eGFR [a] | | | | |
| eGFR <30 mL/min/1.73 m$^2$ | reference | - | - | - |
| eGFR ≥30 mL/min/1.73 m$^2$ | 1.73 (0.67–4.49) | 0.259 | NA | NA |
| Anticoagulant administration Pre-PCI | 0.86 (0.58–1.28) | 0.457 | NA | NA |
| Glycoprotein IIb/IIIa inhibitors administration Peri-PCI | 0.93 (0.53–1.66) | 0.815 | NA | NA |
| Streptokinase dose | | | | |
| 1.5 million units | reference | - | - | - |
| 0.75 million units | 2.82 (1.86–4.27) | <0.001 | 2.82 (1.72–4.61) | <0.001 |
| PCI strategy | | | | |
| rescue PCI | reference | - | - | - |
| routine PCI | 1.52 (0.94–2.44) | 0.086 | NA | NA |
| Number of lesions treated | 0.83 (0.54–1.29) | 0.415 | NA | NA |
| Number of stents implanted | 1.00 (0.79–1.27) | 0.978 | NA | NA |
| intravascular imaging | 4.11 (1.91–8.86) | <0.001 | 5.84 (2.54–13.43) | <0.001 |

[a] classified by major criteria of The Academic Research Consortium for High Bleeding Risk (ARC-HBR) [14], OR, Odds ratio; CI, confidence interval; HT, hypertension; DLP, dyslipidemia; DM, diabetes mellitus; CAD; cardiovascular disease; TIA, transient ischemic attack; eGFR, estimated glomerular filtration rate; PCI, percutaneous coronary intervention; NA, not applicable.

## Discussion

Despite the clinical superiority of ticagrelor compared with clopidogrel among patients with acute coronary syndromes [5], its safety post-fibrinolytic therapy in the ensuing 24 hours remains uncertain. The current study explored the safety and efficacy outcomes of ticagrelor switching versus sustained administered clopidogrel in STE-ACS presenters with pharmacoinvasive strategy of the PCI Center northern Thailand. Overall, during the study period, we found that ticagrelor switching showed a tendency toward increasing from 15% in 2017 to 70% in 2021 of the total annual pharmacoinvasive in STE-ACS population. However, no specific criteria has been established for P2Y$_{12}$ inhibitor selection among patients with STE-ACS post streptokinase; therefore, the matter depends on the physician's judgment. According to our findings, four factors were related to P2Y$_{12}$ inhibitor switching from clopidogrel to ticagrelor in this clinical setting.

The first was patient age; about a 3% chance of receiving ticagrelor decreased with every year of age increase. This was because older age, especially 75 years or more was indicated as a bleeding risk [14], and the TREAT trial [7] excluded these patients. Therefore, patients aged 75 years or more received ticagrelor less than clopidogrel (9.5% and 13.8% in the switch group

**Table 3. Safety and efficacy outcomes.**

| Outcomes | Switch (n = 199) n (%) | No switch (n = 202) n (%) | Unadjusted HR (95% CI) | p-value [a] | Adjusted HR (95% CI) | p-value [b] |
|---|---|---|---|---|---|---|
| **Safety outcomes** | | | | | | |
| Any bleeding | 83 (41.71) | 83 (41.09) | 0.99 (0.73–1.34) | 0.936 | 1.04 (0.74–1.45) | 0.832 |
| BARC type 3 or 5 | 11 (5.53) | 13 (6.44) | 0.84 (0.37–1.87) | 0.662 | 1.31 (0.55–3.10) | 0.545 |
| BARC type 3a to 3c | 10 (5.03) | 13 (6.44) | - | - | - | - |
| BARC type 5 | 1 (0.50) | 0 (0) | - | - | - | - |
| BARC type 1 or 2 | 72 (36.18) | 70 (34.65) | 1.02 (0.73–1.41) | 0.926 | 1.00 (0.69–1.43) | 0.983 |
| **Efficacy outcomes** | | | | | | |
| All-cause death, MI or stroke | 6 (3.02) | 12 (5.94) | 0.48 (0.18–1.28) | 0.141 | 0.57 (0.21–1.57) | 0.279 |
| All-cause death | 2 (1.01) | 7 (3.47) | 0.28 (0.06–1.36) | 0.116 | 0.34 (0.07–1.59) | 0.170 |
| MI | 3 (1.51) | 3 (1.49) | 0.94 (0.19–4.65) | 0.937 | 1.07 (0.20–5.81) | 0.942 |
| Stroke | 1 (0.50) | 2 (0.99) | 0.45 (0.04–4.98) | 0.516 | 0.56 (0.47–6.74) | 0.651 |

[a] p-value for hazard ratio unadjusted, [b] p-value for hazard ratio with IPW adjusted. HR, hazard ratio; CI, confidence interval; BARC, Bleeding Academic Research Consortium; MI, myocardial infarction.

and the no switch group, respectively). The following factors, referring to additional thrombotic risk, included medical history of CAD and use of intravascular imagine, that was reserved for the complex coronary lesion. Thus, from the available evidence, potent $P2Y_{12}$ inhibitors are the standard treatment among patients with ACS [1,2]; patients with high thrombotic risk may provide an opportunity for $P2Y_{12}$ inhibitors switching. The last was receiving 0.75 million units of streptokinase. Because the accelerated streptokinase dose (0.75 million units) was acceptable in both study centers, and the consideration to administer streptokinase regimen was based on the consultant cardiologist's discretion, the current study included about 69.6% of 0.75 million units of streptokinase. However, no difference was noted in bleeding risk between streptokinase 0.75 million units and 1.5 million units (risk ratio 1.05, 95% CI 0.83–1.34, p = 0.671) similar to the ASK-ROMANIA trial [15]. Hence, $P2Y_{12}$ inhibitor switching from clopidogrel to ticagrelor was considered for patients with low bleeding and high thrombotic risk.

The current study revealed the median time from receiving streptokinase to initial administration of ticagrelor was 25.7 hours, which 64% of patients were switched after PCI. While in the TRAET trial [7], the median time from receiving fibrinolysis to initial administration of ticagrelor was 11.4 hours, and Welsh RC et al [8] study was 9.9 hours. The difference in ticagrelor initiation time may depend on their clinical practice and the time to undergo PCI. The Cardiac networks to manage STE-ACS patients in Chiang Mai were set by geographic area. Nakornping Hospital and Maharaj Nakorn Chiang Mai Hospital were the only two PCI Center hospitals. Referral means among this mountainous geographic area is time consumed to these hospitals.

According to our findings, the primary safety outcome did not differ between the switch group and the no switch group. In addition, most of the bleeding events were reported as minor bleeding (BARC type 1 or 2) from PCI access site, and 76% of PCI was performed via the femoral artery. Major bleeding as BARC type 3 or 5 (5.53% and 6.44% in the switch group and no switch group, respectively) was consistent with related studies. The Thai Registry in Acute Coronary Syndrome (TRACS) [9], the nationwide registration, included patients with STE-ACS post fibrinolytic therapy for which most fibrinolytic therapy included streptokinase (94% and 97% in first and second registries, respectively). The in-hospital major bleeding was

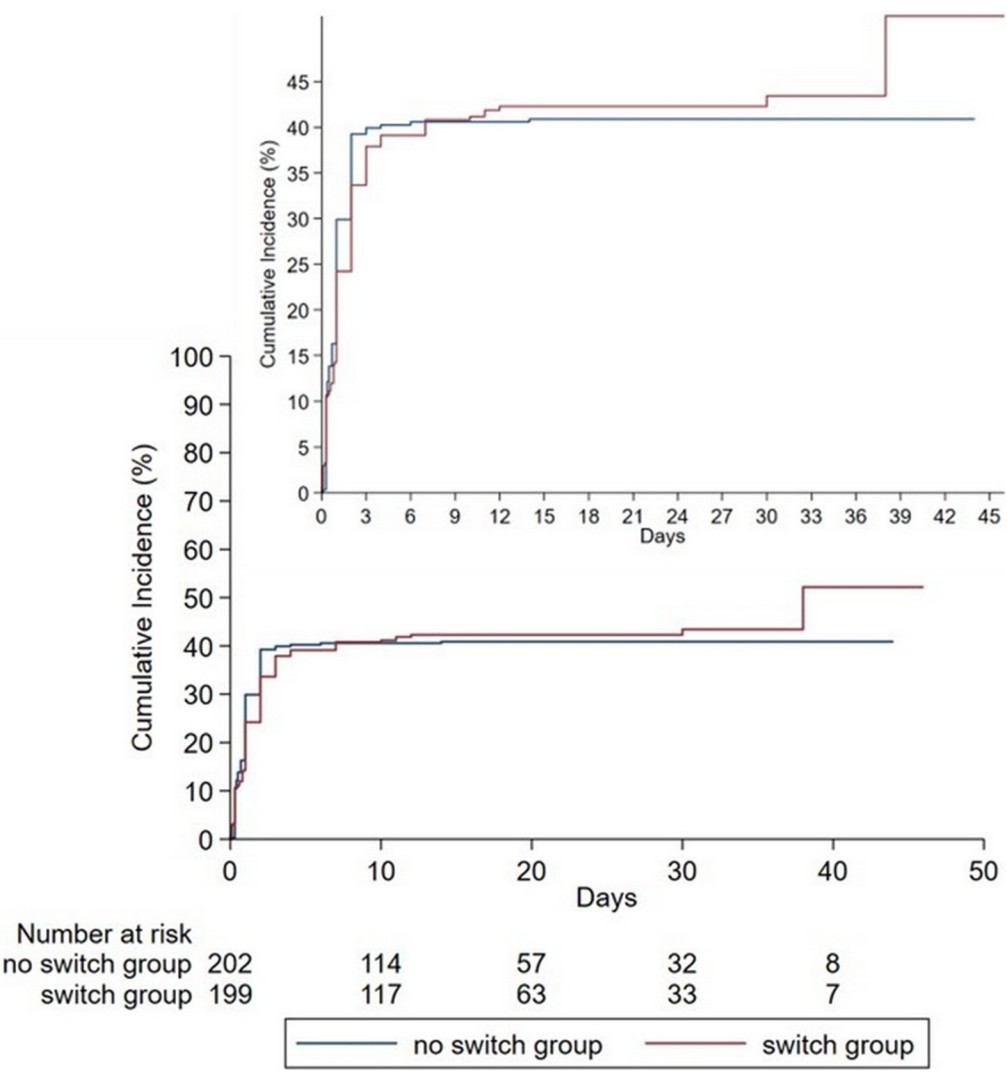

**Fig 2. Kaplan-Meier curve of safety outcomes with adjusted IPW.**

found to be 7.9% and 5.3% in the first and second registries, respectively. However, when compared with the TREAT trial [7], the current study had higher rates of major bleeding because, among patients aged 75 or more, one of the bleeding risks were excluded from the TREAT trial. In addition, the current study demonstrated some different patient characteristics that led to higher bleeding risk. The first was low body weight, the mean body weight was 58.8 ±11.9 kg (Table 1); the weight lower than 65 kg was associated with an increased risk of bleeding [16]. Moreover, patients in the current study had a higher rate of receiving glycoprotein IIb/IIIa inhibitor administration when compared with the TREAT trial, 13.07% and 13.86% in the switch and no switch group, respectively (S1 Table), and 76.3% of patients in the current study were PCI with femoral access, which is associated with a higher risk of bleeding than radial access [17]. Furthermore, approximately 40% of patients in the TREAT trial received fibrin-specific fibrinolysis with tenecteplase, and nearly 20% received non-fibrin-specific agents. Streptokinase is a non-fibrin-specific fibrinolytic, promoting prolongation of prothrombin time for 24 to 48 hours post streptokinase administration and decreases fibrinogen

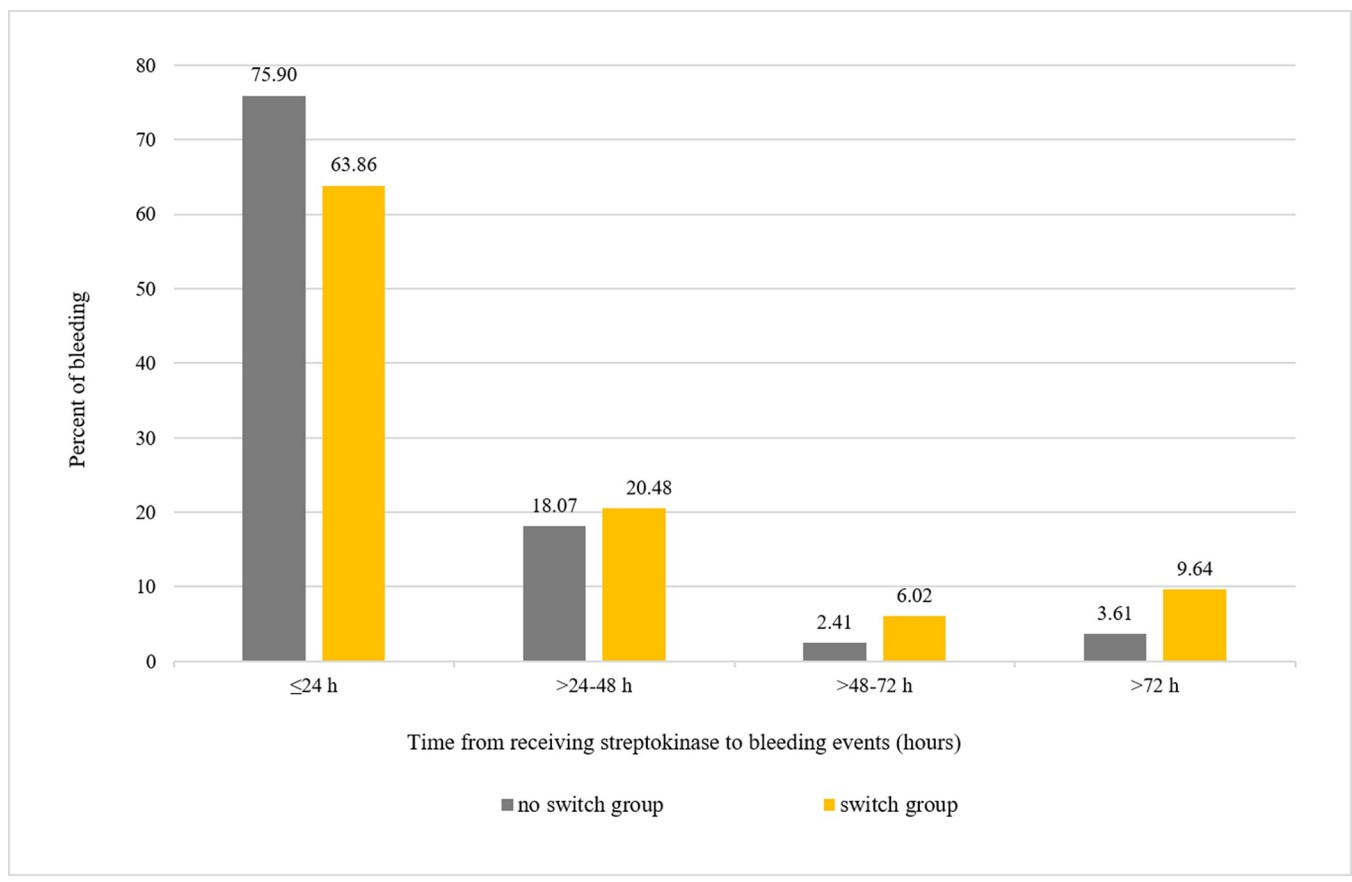

**Fig 3. Bleeding events and time from receiving streptokinase.**

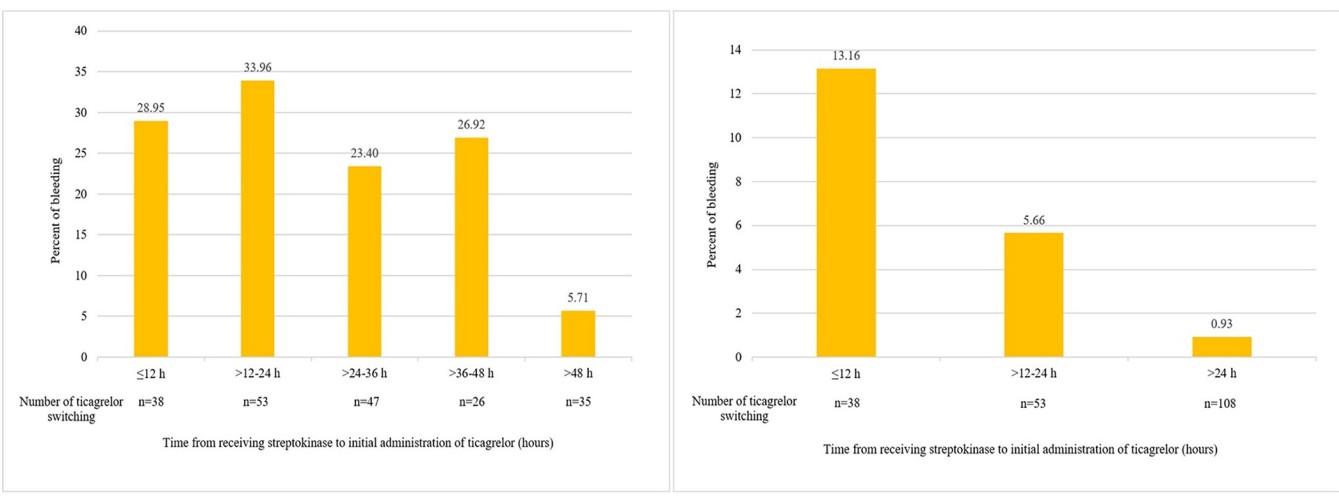

**Fig 4. Bleeding events and time from receiving streptokinase to initial ticagrelor in the switch group, particularly bleeding events post switching.** (a) Any bleeding events; (b) BARC type 3 or 5.

level for 30 hours [18,19]. In addition, Jinatongthai P et al. [20] study showed a higher tendency of major bleeding rate in streptokinase compared with tenecteplase. Therefore, streptokinase may involve a higher bleeding rate than fibrin-specific fibrinolytic, particularly within 24 to 48 hours from streptokinase administration; and thus, bleeding events in the current study exhibited a gradual decrease from the time of streptokinase administration. The other factor that may be related to high bleeding incidence in the current study was ethnicity, which was an independent bleeding predictor. Asians may have a high tendency of bleeding when compared with White or Hispanic ethnicities [21–23].

When considering the bleeding events and ticagrelor switching time from receiving streptokinase in the switch group, showed a high tendency of bleeding with early ticagrelor initiation (Fig 4). According to a variable of the ticagrelor initiation time (median 25.7, IQR 14.6–40.8 hours), the occurrence of bleeding events especially within 24 hours may be confounded with other factors. Also included resulting from medication received during PCI, the median time to PCI was 16.7, IQR 7.2–26.9 hours, particularly parenteral anticoagulants, GP IIb/IIIa additional effect from streptokinase as the above.

Although the switch group showed a higher rate of BARC type 3 or 5 bleeding when early switching, no differences were noted in bleeding events between switching within 12 hours and other times from streptokinase administration (S1 Fig). In the TREAT trial [7], the median time of randomization was 11.4 h after fibrinolysis with antiplatelet study medications administered thereafter, showing non-inferiority of bleeding between ticagrelor and clopidogrel, agreeing with the current study.

The secondary composite outcomes including all-cause death, MI or stroke showed a non-difference between the switch group and the no switch group. However, when compared with the Study of Platelet Inhibition and Patient Outcomes (PLATO) [5], in which the composite outcomes occurred in 9.8% of patients receiving ticagrelor as compared with 11.7% of those receiving clopidogrel, our study showed a lower rate of composite efficacy outcomes due to the difference in reperfusion strategy and a low number of patients in the cohort. As a result, the comparison of efficacy outcomes rate and statistical power to evaluate superiority was limited. Nevertheless, the composite efficacy outcomes in the current study are similar to the TREAT trial [7] and that of Welsh RC et al. [8] study, which included only patients with pharmacoinvasive strategy similar to the current study.

Several limitations were encountered in this study. First, the study employed a retrospective observational design. Although the outcomes were computed with adjusted IPW; the unknown confounding factor might have persisted. Second, the composite of efficacy outcomes may have been underestimated due to the short follow-up time and small number of cohort patients enrolled. Thus, another study with a large number of patients and extended study time may be considered. However, our study possessed several potential strengths. The current study exhibited the actual clinical practice among patients with STE-ACS with pharmacoinvasive strategy of two PCI Centers of northern Thailand. In addition, our study could be a part of data support system to decrease the gap of evidence in $P2Y_{12}$ inhibitor switching after fibrinolysis with streptokinase and assist physicians in selecting $P2Y_{12}$ inhibitors in this clinical setting.

## Conclusion

The study indicated a pattern of $P2Y_{12}$ inhibitor selection for patients with STE-ACS post streptokinase therapy in real-world practice in Thailand. Our data supported that ticagrelor switching post streptokinase therapy did not significantly differ regarding any bleeding events and BARC type 3 or 5 when compared with clopidogrel.

## Supporting information

**S1 Checklist. STROBE Statement—checklist of items that should be included in reports of observational studies.**
(PDF)

**S1 Fig. Bleeding events and time from streptokinase to ticagrelor initiation in the switch group, particularly bleeding events post switching.**
(PDF)

**S1 Table. Procedures and other baseline characteristics.**
(PDF)

**S2 Table. Standardized difference of baseline characteristics with and without adjusted IPW.**
(PDF)

## Acknowledgments

We would like to acknowledge Assoc. Prof. Surakit Nathisuwan, Assoc. Prof. Surarong Chin-wong and Assoc. Prof. Dujrudee Chinwong for their statistical and method suggestion.

## Author Contributions

**Conceptualization:** Phornpaka Ueapornpanith, Voratima Yoodee.

**Data curation:** Phornpaka Ueapornpanith, Boonyanuch Buranakiti, Thanyalak Chotayaporn.

**Formal analysis:** Phornpaka Ueapornpanith.

**Methodology:** Phornpaka Ueapornpanith, Arintaya Phrommintikul, Voratima Yoodee.

**Writing – original draft:** Phornpaka Ueapornpanith.

**Writing – review & editing:** Arintaya Phrommintikul, Voratima Yoodee.

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
