## [Decision Letter · Decision Letter 0]

18 Jun 2023

PONE-D-23-10765Safety Outcomes of Ticagrelor among patients with STE-ACS post streptokinase therapy-a Retrospective Observational StudyPLOS ONE

Dear Dr. Yoodee,

Thank you for submitting your manuscript to PLOS ONE. After careful consideration, we feel that it has merit but does not fully meet PLOS ONE’s publication criteria as it currently stands. Therefore, we invite you to submit a revised version of the manuscript that addresses the points raised during the review process.

The topic is interesting, though trombolysis is rarely used, just in countries where PCI is not available 24h/7d. We suggst the Authors to highligh reasons for bleeding events other than thrombolysis. The variable time of ticagrelor administration should be further discussed. 

We look forward to receiving your revised manuscript.

Kind regards,

Chiara Lazzeri

Academic Editor

PLOS ONE

Journal Requirements:

Reviewers' comments:

Reviewer's Responses to Questions

**Comments to the Author**

1. Is the manuscript technically sound, and do the data support the conclusions?

Reviewer #1: Partly

2. Has the statistical analysis been performed appropriately and rigorously? 

Reviewer #1: No

3. Have the authors made all data underlying the findings in their manuscript fully available?

Reviewer #1: Yes

4. Is the manuscript presented in an intelligible fashion and written in standard English?

Reviewer #1: Yes

5. Review Comments to the Author

Reviewer #1: Τhe topic of the article is interest and World cardiology community has to consider seriously the thrombolytic therapy as an alternative therapy when Primary PCI cannot be performed in patients with STEMI. Indeed, thrombolysis is used frequently in countries and regions where there is no 24/7 PPCI Service.

After reading the proposed article a have to make some considerations.

1.Streptokinase is rarely used now days in Europe and USA so there is no interest in the western countries about its use but i understand that it is still used in some countries and it has to be studied.

2.The bleeding events, any bleeding and major bleeding, were much higher compared to other relative studies (TREAT trial) and I am not convinced that this fact is due only to the use of streptokinase as the authors explain.

3.The population of the switch group had a greater percentage of medical history of coronary artery disease and probably they were already receiving an antiplatelet agent that could increase the bleeding risk, i think this question merits to be clarified.

4.The timing of ticagrelor administration after thrombolysis was very variable in the switch group so the effect of ticagrelor on bleeding it is not uniform in time. Most of bleeding events occurred within 24 h after thrombolysis when 91 out of 199 patients had received ticagrelor according fig. 4.

5. I would like to see a statistical calculation justifying the 401 patients studied as a sample size large enough in order to detect as significant bleeding events differences between the two groups, according previous relative studies.

6. PLOS authors have the option to publish the peer review history of their article (what does this mean?). If published, this will include your full peer review and any attached files.

Reviewer #1: No

---

## [Author Response · Author response to Decision Letter 0]

19 Jul 2023

Reviewer 1

Comments to the Author 

1. Streptokinase is rarely used now days in Europe and USA so there is no interest in the western countries about its use but I understand that it is still used in some countries and it has to be studied.

Author response: We appreciate the Reviewer’s input to review our work.

2. The bleeding events, any bleeding and major bleeding, were much higher compared to other relative studies (TREAT trial) and I am not convinced that this fact is due only to the use of streptokinase as the authors explain.

Author response: Thank you for your valuable pointing. We have realized the importance of additional discussion about the difference in major bleeding between the current study and the TREAT trial and we have implemented this change according to your suggestions to complete and provide more information to the context. Here are the original and revised version:

 Original version: However, when compared with the TREAT trial[7], the current study had higher rates of major bleeding because, among patients aged 75 or more, one of the bleeding risks were excluded. Furthermore, approximately 40% of patients in the TREAT trial received fibrin-specific fibrinolysis with tenecteplase, and nearly 20% received non-fibrin-specific agents. Streptokinase is a non-fibrin-specific fibrinolytic, promoting prolongation of prothrombin time for 24 to 48 hours post streptokinase administration and decreases fibrinogen level for 30 hours[16, 17]. In addition, Jinatongthai P et al[18] study showed a higher tendency of major bleeding rate in streptokinase compared with tenecteplase. Therefore, streptokinase may involve a higher bleeding rate than fibrin-specific fibrinolytic, particularly within 24 to 48 hours from streptokinase administration; and thus, bleeding events in the current study exhibited a gradual decrease from the time of streptokinase administration.

Revised version: However, when compared with the TREAT trial[7], the current study had higher rate of major bleeding because among patients aged 75 or more, one of the bleeding risks was excluded from the TREAT trial. In addition, the current study demonstrated some different patient characteristics that led to higher bleeding risk. The first was low body weight, the mean body weight was 58.8±11.9 kg (Table 1); the weight lower than 65 kg was associated with an increased risk of bleeding[16]. Moreover, patients in the current study had a higher rate of receiving glycoprotein IIb/IIIa inhibitor administration when compared with the TREAT trial, 13.07% and 13.86% in the switch and no switch group, respectively (S1 Table), and 76.3% of patients in the current study were PCI with femoral access, which is associated with a higher risk of bleeding than radial access [17]. Furthermore, approximately 40% of patients in the TREAT trial received fibrin-specific fibrinolysis with tenecteplase, and nearly 20% received non-fibrin-specific agents. Streptokinase is a non-fibrin-specific fibrinolytic, promoting prolongation of prothrombin time for 24 to 48 hours post streptokinase administration and decreases fibrinogen level for 30 hours[18, 19]. In addition, Jinatongthai P et al[20] study showed a higher tendency of major bleeding rate in streptokinase compared with tenecteplase. Therefore, streptokinase may involve a higher bleeding rate than fibrin-specific fibrinolytic, particularly within 24 to 48 hours from streptokinase administration; and thus, bleeding events in the current study exhibited a gradual decrease from the time of streptokinase administration. The other factor that may be related to high bleeding incidence in the current study was ethnicity, which was an independent bleeding predictor. Asians may have a high tendency of bleeding when compared with White or Hispanic ethnicities [21-23].

(Location: section “Discussion” Lines 222-228 and 237-239, page 14-15).

3. The population of the switch group had a greater percentage of medical history of coronary artery disease and probably they were already receiving an antiplatelet agent that could increase the bleeding risk, I think this question merits to be clarified.

Author response: Thank you for your valuable pointing. In order to answer your question, we would like to refer from Cho JY, et al. study, the post hoc analysis of the TICO trial, demonstrating factors related to bleeding. From the study showed no significant difference in major bleeding between patients with the presence or absence of prior PCI. Also, in the PLATO trial, there was no interaction of the major bleeding when compared between patients with medical history of MI and without MI group. There was no trend of bleeding increasing among patients receiving antiplatelet previously. However, the above resulted from subgroup analysis with a low number of patients. Thus, we realized that to provide more appropriate confounding factors controlling, the RCT should be performed in the future. 

4. The timing of ticagrelor administration after thrombolysis was very variable in the switch group so the effect of ticagrelor on bleeding it is not uniform in time. Most of bleeding events occurred within 24 h after thrombolysis when 91 out of 199 patients had received ticagrelor according fig. 4.

Author response: Thank you for your valuable pointing. In Fig 4 we demonstrate bleeding events and switching time in the switch group that included only bleeding events after the switching. The number of patients according to the x-axis means the total number of patients who initiated ticagrelor at that time frame and the percentages of bleeding in the y-axis were calculated from the number of patients with bleeding events and the number of switching at that time. As you have raised this point, we have realized the importance of additional discussion. We have implemented this change according to your suggestions to complete and provide more information to the context. Here is the additional context in the revised version.

 Revised version: When considering the bleeding events and ticagrelor switching time from receiving streptokinase in the switch group, showed a high tendency of bleeding with early ticagrelor initiation (Fig 4). According to a variable of the ticagrelor initiation time (median 25.7, IQR 14.6-40.8 hours), the occurrence of bleeding events especially within 24 hours may be confounded with other factors. Also included resulting from medication received during PCI, the median time to PCI was 16.7, IQR 7.2-26.9 hours, particularly parenteral anticoagulants, GP IIb/IIIa additional effect from streptokinase as the above.

(Location: section “Discussion” Lines 240-245, page 15 and “Fig 4”).

5. I would like to see a statistical calculation justifying the 401 patients studied as a sample size large enough in order to detect as significant bleeding events differences between the two groups, according previous relative studies.

Author response: Thank you for your time and consideration, as well as the invaluable comments and suggestions. We performed sample size and power calculations based on a superiority comparison with 80% power and two side alpha of 5%. From Table 1, we expected the any bleeding rate of 23.4%. Because based on the study among patients with pharmacoinvasive strategy with SK and also received guideline recommendation medication including DAPT that may exhibit the real practices management in the setting using SK. We assumed that any bleeding difference was 41% (Table 2), so the estimated any bleeding in the switch group was 33.0%. As a result, the current study will require 344 patients in each group. The formula for sample size calculation was described below and the sample size estimation was presented in Table 3. 

All the tables will be presented in the attached file.

---

## [Editor Report · Decision Letter 1]

24 Jul 2023

Safety Outcomes of Ticagrelor among patients with STE-ACS post streptokinase therapy-a Retrospective Observational Study

PONE-D-23-10765R1

Dear Dr. Yoodee,

We’re pleased to inform you that your manuscript has been judged scientifically suitable for publication and will be formally accepted for publication once it meets all outstanding technical requirements.

Kind regards,

Chiara Lazzeri

Academic Editor

PLOS ONE